# EXPERT-DATA ALIGNMENT GOVERNS GENERATION QUALITY IN DECENTRALIZED DIFFUSION MODELS

**Marcos Villagra, Bidhan Roy, Raihan Seraj & Zhiying Jiang**
Bagel Labs (`www.bagel.com`)
Schrodinger Journey Labs, Inc. (dba Bagel Labs), 251 Little Falls Drive
Wilmington, DE 19808, United States
{`marcos,bidhan,raihan,gin`}`@bagel.com`

## ABSTRACT

Decentralized Diffusion Models (DDMs) route denoising through experts trained independently on disjoint data clusters, which can strongly disagree in their predictions. What governs the quality of generations in such systems? We present the first systematic investigation of this question. A priori, one might expect that minimizing denoising trajectory sensitivity should govern generation quality. We demonstrate this hypothesis is incorrect: full ensemble routing achieves the most stable sampling dynamics while producing the worst generation quality (FID 47.9 vs. 22.6 for sparse Top-2 routing). Instead, we identify *expert-data alignment* as the governing principle: generation quality depends on routing inputs to experts whose training distribution covers the current denoising state. Across two DDM systems, we validate this principle using cluster distance analysis, per-expert prediction quality, and expert disagreement analysis. For DDM deployment, our findings establish that routing should prioritize expert-data alignment over numerical stability metrics.

## 1 INTRODUCTION

Decentralized Diffusion Models (DDMs) (McAllister et al., 2025) combine independently trained diffusion experts (Ho et al., 2020) via an inference-time router. Because experts are trained on disjoint data clusters and can strongly disagree in their predictions, understanding what governs generation quality becomes crucial—yet this question has not been systematically studied.

A natural hypothesis is that numerical stability determines quality (Yang et al., 2023): routing strategies that minimize trajectory sensitivity should produce superior samples. **We demonstrate this hypothesis is incorrect.** Full ensemble routing, which combines all expert predictions at each step, achieves the lowest trajectory sensitivity yet produces the worst generation quality (Table 1).

We identify *expert-data alignment*—routing inputs to experts trained on similar data—as the governing principle. When sparse routing (e.g., Top-2) selects experts whose training distribution covers the current denoising state, each expert produces coherent velocity predictions that combine meaningfully. Full ensemble routing forces all experts to process every input; since each expert is trained on only a subset of the data, most process out-of-distribution data at any given time. The averaged velocity field may be smooth, but it points toward an incoherent compromise rather than the data manifold.

We provide direct experimental validation of this principle across two distinct DDM systems. Data-cluster distance analysis confirms that sparse routing selects experts with data clusters closest to the input embedding. Per-expert prediction quality analysis shows that selected experts produce velocity predictions with higher alignment to the blended output. Expert disagreement analysis demonstrates that disagreement under full ensemble correlates with quality degradation.

**Contributions.** (1) We identify expert-data alignment as the primary determinant of generation quality in DDMs, validated through cluster distance analysis, per-expert prediction quality, and expert disagreement analysis. (2) We demonstrate a stability–quality dissociation: trajectory sensitivity does not govern generation quality. Full details appear in the Appendix.

Table 1: **Stability–quality dissociation.** Full ensemble achieves the lowest trajectory sensitivity ($\widehat{L}_{\text{eff}}^{(h)}$) and step-refinement disagreement ($\Delta_{\text{refine}}$), yet produces the worst FID. FID from Jiang et al. (2025); other metrics measured on $n=1000$ samples.

| Strategy | FID $\downarrow$ | $\widehat{L}_{\text{eff}}^{(h)}$ | $\Delta_{\text{refine}} \downarrow$ |
|---|---|---|---|
| Top-1 | 30.60 | 18.81 $\pm 7.15$ | 0.075 $\pm 0.107$ |
| Top-2 | **22.60** | 17.48 $\pm 6.07$ | 0.051 $\pm 0.070$ |
| Full (8) | 47.89 | **17.07** $\pm 6.33$ | **0.020** $\pm 0.020$ |

## 2 BACKGROUND

**Diffusion sampling.** Diffusion sampling generates data by integrating an ordinary differential equation (ODE) $\frac{dx_t}{dt} = v_t(x_t)$ from noise to data (Lipman et al., 2022). We call $v = \{v_t\}_{t=0}^1$ a *flow* and the solution $\{x_t\}$ a *trajectory*. The ODE has a unique solution when $v$ is Lipschitz continuous (Coddington et al., 1956).

**Decentralized diffusion.** DDMs (McAllister et al., 2025) train $K$ expert diffusion models in complete isolation on disjoint data partitions. At inference, a lightweight router predicts weights $w_t^{(k)}(x_t) \geq 0$ with $\sum_k w_t^{(k)} = 1$ at each denoising step. The routed velocity field is $v_t(x_t) = \sum_{k=1}^K w_t^{(k)}(x_t) v_t^{(k)}(x_t)$. Sampling integrates $\frac{dx_t}{dt} = v_t(x_t)$ from noise $x_0 \sim \mathcal{N}(0, I)$. Top-$k$ routing selects the $k$ experts with highest probability and renormalizes their weights (Shazeer et al., 2017). Full ensemble uses all experts weighted by router probabilities.

Unlike traditional MoE where experts are FFN layers within a shared backbone trained jointly with load balancing losses, DDM experts are *complete diffusion models* trained in isolation on disjoint data partitions—no shared parameters, no gradient communication, no joint training. Routing occurs at the *input level* (entire noisy images) rather than token level, and experts are combined only at inference time. This distinction matters because DDM experts, having never coordinated during training, can produce arbitrarily different outputs for the same input.

## 3 THE STABILITY–QUALITY DISSOCIATION

A natural hypothesis is that numerical stability governs generation quality in DDMs where routing strategies that minimize trajectory sensitivity should produce superior samples. Classical stability analysis suggests DDMs should fail—Lipschitz constants of deep networks grow exponentially with depth (Fazlyab et al., 2019; Virmaux & Scaman, 2018), and Grönwall's inequality implies small perturbations amplify over integration (Hairer et al., 1993). We establish that this hypothesis is incorrect.

We measure trajectory-local sensitivity via $\widehat{L}_{\text{eff}}^{(h)} := \max_n \|J_x v(\tilde{x}_{t_n}, t_n)\|$, the maximum Jacobian spectral norm along the numerical trajectory. We also measure step-refinement disagreement $\Delta_{\text{refine}}$ defined as the LPIPS distance (Zhang et al., 2018) between samples generated with $N$ and $2N$ solver steps from the same initial noise, providing a methodologically independent measure of numerical convergence.

Table 1 shows the key result where full ensemble achieves the lowest $\widehat{L}_{\text{eff}}^{(h)}$ and $\Delta_{\text{refine}}$, yet produces the worst FID—even worse than a single expert (Top-1). Top-2 achieves the best FID despite higher trajectory sensitivity. This rules out numerical stability as the quality determinant. The monolithic baseline (a single model trained on all data) achieves FID 29.64, showing that Top-2 routing can outperform centralized training while using experts trained in complete isolation.

Table 2: **Cluster distance analysis validates expert-data alignment.** Lower mean rank = better alignment (1=closest). Results averaged over $n = 500$ samples.

| Routing | Mean Rank $\downarrow$ | Top-2 Match $\uparrow$ |
|---------|------------------------|------------------------|
| Top-1   | $1.54 \pm 0.28$        | 90.2%                  |
| Top-2   | $1.96 \pm 0.26$        | 83.9%                  |
| Full (8)| $4.50 \pm 0.00$        | 25.0%                  |

## 4 EXPERT-DATA ALIGNMENT

### 4.1 THE HYPOTHESIS

The preceding section ruled out numerical stability as the quality determinant. We now establish *expert-data alignment*—routing inputs to experts trained on similar data—as the governing principle.

Full ensemble averaging produces a smoother velocity field because averaging over all experts reduces variance in velocity predictions. However, this smoothing forces experts to process out-of-distribution data. Each expert is trained on only one cluster; when all experts contribute to every input, most process data outside their training distribution. The averaged velocity may be smooth but points toward an incoherent compromise rather than the data manifold. Top-2 routing selects experts whose training data matches the current input, keeping each expert producing coherent velocity predictions that combine meaningfully.

If expert-data alignment governs quality, we expect: (1) sparse routing should select experts with lower cluster distance; (2) selected experts should produce superior velocity predictions; (3) expert disagreement should correlate with quality degradation.

### 4.2 EXPERIMENTAL VALIDATION

We use the pretrained DDM Paris model (Jiang et al., 2025) with $K = 8$ experts trained on a subset of LAION-Aesthetics (Schuhmann et al., 2022). The dataset was partitioned into 8 semantic clusters via two-stage hierarchical k-means on DINOv2-ViT-L/14 embeddings (Oquab et al., 2023). The model uses a DiT-B/2 router ($\sim$129M parameters) and 8 DiT-XL/2 experts ($\sim$606M parameters each, $\sim$5B total).

**Cluster distance analysis.** Let $\mathcal{C}_k$ denote the training data cluster for expert $k$, and let $d(x, \mathcal{C}_k)$ denote the Euclidean distance from the DINOv2 embedding of input $x$ to the centroid of cluster $\mathcal{C}_k$. We define *high expert-data alignment* as the condition where selected experts have low $d(x, \mathcal{C}_k)$ relative to non-selected experts. For $n = 500$ samples, we extract DINOv2 embeddings at timesteps $t \in \{0.3, 0.5, 0.7\}$ and compute the distance to each of the 8 cluster centroids. Table 2 shows that Top-1 and Top-2 achieve mean cluster ranks of 1.54 and 1.96 (1=closest), far below the 4.5 baseline of full ensemble, with Top-2 match rates exceeding 83%.

**Per-expert prediction quality.** For $n = 200$ samples with Top-2 routing, we record individual velocity predictions $v_t^{(k)}(x_t)$ for all 8 experts and the blended velocity $v_t(x_t) = \sum_{k \in \mathcal{S}} w_t^{(k)} v_t^{(k)}(x_t)$. We compute the velocity alignment score as the cosine similarity between each expert's prediction and the blended output, reporting angular deviation in degrees. Selected experts achieve 3.6° deviation vs. 5.1° for non-selected experts (independent samples $t$-test, $p < 0.001$), a 29% reduction confirming systematic identification of coherent experts.

**Expert disagreement and quality.** For $n = 500$ full ensemble samples, we measure mean pairwise expert disagreement $D(x_t) = \frac{1}{\binom{K}{2}} \sum_{i<j} \|v_i(x_t) - v_j(x_t)\|_2$ and trajectory-integrated disagreement $D_{\text{int}} = \int_0^1 D(x_t)\,dt$. We compare perceptual quality via LPIPS distance to corresponding Top-2 outputs (matched initial noise). Figure 1 shows monotonic increase in LPIPS across disagreement quartiles, confirming that expert disagreement drives quality degradation. This correlation provides causal evidence: high disagreement represents states where alignment naturally breaks down.

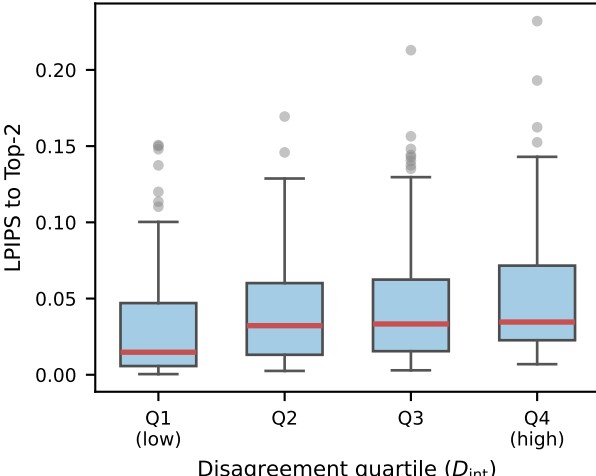

Figure 1: **Expert disagreement degrades quality.** Samples binned by trajectory-integrated disagreement (Q1=lowest, Q4=highest). Higher disagreement $\rightarrow$ greater LPIPS distance from Top-2 reference.

**MNIST validation.** We validate on a separate MNIST DDM with 10 UNet experts (each $\sim$10M parameters) trained on digit-specific subsets, creating strong expert specialization. The alignment effects are even more pronounced where selected experts show 43% lower angular deviation (6.4° vs. 11.3°, $p \approx 0$). The larger gap compared to Paris DDM (4.9° vs. 1.5°) reflects stronger specialization on digit-specific subsets. Under full ensemble routing, trajectory-integrated disagreement correlates even more strongly with quality degradation. Details in Appendix B.

## 5 Discussion and Conclusions

Our central finding is that expert-data alignment governs generation quality in DDMs, not numerical stability. The experimental results reveal a fundamental tradeoff where strategies that maximize numerical stability (full ensemble) necessarily sacrifice expert-data alignment, and vice versa.

Cluster distance analysis quantifies this directly—sparse routing achieves mean cluster ranks of 1.54–1.96 versus 4.50 for full ensemble (Table 2)—while per-expert analysis shows selected experts produce 29% lower angular deviation from the blended velocity. The disagreement-quality correlation (Figure 1) provides causal evidence, that is, when alignment breaks down, sample quality degrades proportionally. Critically, high disagreement represents states where multiple experts are forced to process inputs outside their training distribution.

Furthermore, trajectory-local sensitivity ($\widehat{L}_{\mathrm{eff}}^{(h)}$) does not predict quality across routing strategies, but may serve as a within-strategy diagnostic for identifying numerically sensitive trajectories. The weak correlation $\rho(\widehat{L}_{\mathrm{eff}}^{(h)}, \Delta_{\mathrm{refine}}) < 0.08$ suggests factors beyond worst-case Jacobian norms—such as directional alignment of perturbations with the flow or error cancellation across timesteps—affect actual error accumulation.

For practitioners deploying DDMs with independently trained experts, our findings demonstrate that routing which maintains expert-data alignment is more important than optimizing for numerical stability metrics. Future work should explore training objectives that improve expert robustness to out-of-distribution inputs. Detailed analysis including convergence arguments, additional experiments, and limitations appears in the Appendix.

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

## APPENDIX

## A   RELATED WORK

**Diffusion as ODE/SDE and numerical stability.** Diffusion sampling involves Lipschitz-constrained ODEs where discretization error affects accuracy. Sampling can be expressed as a probability-flow Ordinary Differential Equation (ODE) Song et al. (2020), where Lipschitz constants and discretization error determine solver accuracy. Recent work by Tan et al. (2025) addresses temporal stiffness via a specialized solver called STORK based on stabilized Runge-Kutta methods for stiff diffusion ODEs. STORK's stabilized solvers could, in principle, be combined with our approach (using stable solvers for temporal stiffness while using sparse routing to control spatial sensitivity). We use the released Euler and Heun solvers to isolate the effect of routing strategies, but STORK-style solvers may provide additional gains in decentralized settings.

**Local Lipschitz analysis.** Local Lipschitz bounds have been extensively studied in the neural network literature. Jordan & Dimakis (2020) developed methods for exactly computing local Lipschitz constants, enabling input-specific sensitivity analysis rather than global worst-case bounds. Our work applies this trajectory-local perspective to diffusion sampling. The key novelty is not the local Lipschitz concept itself, but its application to understanding decentralized diffusion dynamics and the discovery that routing implicitly stabilizes the sampling dynamics.

**Decentralized diffusion.** DDMs show that decentralized experts, when routed, can match a monolithic diffusion objective McAllister et al. (2025). Our work investigates why such combination succeeds despite expert disagreement, providing a stability-based explanation complementary to the original capacity arguments.

**DDM vs. traditional Mixture-of-Experts.** DDMs are *ensembles of independently trained models*, not traditional Mixture-of-Experts (MoE). In standard MoE architectures experts are FFN layers within a shared backbone, trained jointly with load balancing losses, and routed at the token level Shazeer et al. (2017); Fedus et al. (2022). In DDM, each "expert" is a *complete diffusion model* trained in isolation on a disjoint data partition (no shared parameters, no gradient communication, no joint training). Routing occurs at the *input level* (entire noisy images) rather than token level, and experts are combined only at inference time. Concurrent work applies traditional MoE architectures *within* diffusion models Fei et al. (2024); Sun et al. (2024); Shi et al. (2025); Yuan et al. (2025); Cheng et al. (2025); Liu et al. (2025); Zheng et al. (2025); DDM instead combines complete, independently trained models. This distinction matters for stability analysis: DDM experts can produce arbitrarily different outputs for the same input (having never coordinated during training), whereas MoE experts share a representational backbone that constrains their disagreement. Our analysis specifically addresses the stability challenges arising from DDM's decentralized training.

**Data-aware routing and expert specialization.** The importance of matching inputs to appropriately trained experts is well-established in MoE systems. Sparsely-gated MoE architectures rely on learned routing to direct inputs to relevant experts Shazeer et al. (2017), with subsequent work analyzing expert utilization and load balancing Dai et al. (2022); Zhou et al. (2022). In federated learning, data heterogeneity across clients creates analogous challenges: models trained on non-IID partitions may produce poor predictions on out-of-distribution inputs Li et al. (2020). Our work provides direct experimental evidence that this principle governs sample quality in DDMs: sparse routing succeeds precisely because it maintains alignment between inputs and expert training distributions.

## B  MNIST VALIDATION OF EXPERT-DATA ALIGNMENT

We validate the expert-data alignment hypothesis on a separate MNIST-based DDM with 10 UNet experts and a CNN router. This controlled setting provides independent confirmation of our main findings on a simpler domain with more specialized experts.

The MNIST DDM consists of 10 independently trained UNet experts (each $\sim$10M parameters) and a lightweight CNN router. Each expert was trained on a digit-specific subset of MNIST, creating strong expert specialization. The router was trained separately to predict which expert best matches each input.

We run two experiments mirroring the Paris DDM analysis: (1) per-expert prediction quality comparing selected vs. non-selected experts, and (2) expert disagreement correlation with quality degradation. All experiments use the Heun solver with 50 steps and $n = 500$ samples per configuration.

Table 3 shows that selected experts produce smaller angular deviation from the blended velocity than non-selected experts. Selected experts achieve smaller angular deviation from the blended velocity ($6.4°$ vs. $11.3°$, $p \approx 0$), a 43% reduction confirming systematic identification of coherent experts. The angular deviation gap ($4.9°$) is substantially larger than Paris DDM ($1.5°$), reflecting the stronger specialization of MNIST experts trained on digit-specific subsets.

Under full ensemble routing, we measure the correlation between trajectory-integrated expert disagreement and output quality degradation (MSE and LPIPS distance to Top-2 reference outputs). MNIST exhibits a substantially stronger disagreement-quality correlation than Paris DDM. This

Table 3: **Selected experts produce better-aligned predictions.** Angular deviation (degrees) from blended velocity for selected versus non-selected experts under Top-2 routing. Smaller is better. Results averaged over $n = 500$ samples.

| Expert Status | Angular Dev. ↓ | Std Dev |
|---|---|---|
| Selected (Top-2) | 6.4° | ±1° |
| Non-selected | 11.3° | ±1° |
| Reduction | 4.9° (43%) | $p \approx 0$ |

stronger correlation confirms that expert disagreement is a robust predictor of quality degradation, with the effect amplified in settings with stronger expert specialization.

## C  TRAJECTORY SENSITIVITY ANALYSIS

Having established that expert-data alignment governs sample quality, we now analyze trajectory sensitivity to understand when numerical convergence holds and to develop within-strategy diagnostics. We present a conditional convergence argument linking trajectory-local sensitivity to DDM convergence, followed by empirical validation.

### C.1  CONVERGENCE IN PROBABILITY UNDER TRAJECTORY-LOCAL SENSITIVITY

We present a conditional argument where sensitivity bounds around a trajectory imply convergence in probability of the DDM sampler.

**Definition C.1** (Trajectory-local sensitivity). Given a flow $v$ and an initial condition $x_1$, let $\{x_t\}_{t\in[0,1]}$ denote the (exact) ODE solution for $t \in [0,1]$. If the solution does not exist over $[0,1]$, we let $L_{\text{eff}}(x_1) = +\infty$. Otherwise, we define the *effective Lipschitz constant* at $x_1$ as $L_{\text{eff}}(x_1) = \sup_{t\in[0,1]} \|J_x v_t(x_t)\|$. Here, $\| \cdot \|$ denotes the operator norm induced by the Euclidean norm (i.e., the Jacobian spectral norm). We call the trajectory *locally stable* if $L_{\text{eff}}(x_1) < \infty$.

Definition C.1 introduces $L_{\text{eff}}(x_1)$ as a functional of the exact ODE solution $\{x_t\}$ induced by $(v, x_1)$. This is not a dynamical claim about where trajectories go, and we do not use it to prove that trajectories enter or remain in low-sensitivity regions. Instead, boundedness of $L_{\text{eff}}$ is an *assumption* in our numerical convergence argument, which we support empirically via a numerical approximate proxy $\widehat{L}_{\text{eff}}^{(h)}$ defined below.

**Definition C.2.** Given a numerical solver with step size $h$ producing a discrete trajectory $\{\tilde{x}_{t_n}^{(h)}\}_{n=0}^N$, we define the *empirical effective Lipschitz constant* as $\widehat{L}_{\text{eff}}^{(h)}(x_1) := \max_{n\in\{0,...,N\}} \|J_x v(\tilde{x}_{t_n}^{(h)}, t_n)\|$.

The numerical approximation $\widehat{L}_{\text{eff}}^{(h)}$ is a good estimator of $L_{\text{eff}}$ as we refine the solver (see Appendix D for details), and hence, it can be used as a post-hoc diagnostic.

*Remark* C.3 (Circularity of the diagnostic). Computing $\widehat{L}_{\text{eff}}^{(h)}$ requires the numerical trajectory, which is only available *after* sampling completes. This makes $\widehat{L}_{\text{eff}}^{(h)}$ a retrospective diagnostic rather than an a priori predictor: one cannot use it to decide whether a given initial noise will yield a well-converged sample without first running the sampler. We view this as inherent to trajectory-local analysis and use $\widehat{L}_{\text{eff}}^{(h)}$ primarily for understanding sensitivity differences across routing strategies, not for sample-level prediction.

**Definition C.4** (Sampler sensitivity). A sampler is $(L, \delta)$-*trajectory-locally sensitive* if $P_{x_1\sim q}[L_{\text{eff}}(x_1) \leq L] \geq 1 - \delta$, for some noise distribution $q$, initial condition $x_1$, and constants $L$ and $\delta$.

Consider a one-step numerical ODE solver with step size $h$ (e.g., Heun / explicit trapezoidal rule) approximating the initial value problem $\frac{dx_t}{dt} = v_t(x_t)$ on $t \in [0,1]$, with discrete time steps $t_n = 1 - nh$ for $n = 0, \ldots, N$, where $h = 1/N$ and $N$ is the total number of steps. At each step $n$, the method has a (one-step) local truncation error $\epsilon_n^{\text{local}}$ defined as $\epsilon_n^{\text{local}}(t_n) = x_{t_{n+1}} - \text{Step}_h(x_{t_n}, t_n)$,

where $\text{Step}_h(x,t)$ denotes the numerical update. Let $e_N := \|x_{t_N} - \tilde{x}_{t_N}^{(h)}\|$ denote the global solution error at the final step. A standard Grönwall-based argument shows that on the event $\{L_{\text{eff}}(x_1) \leq L\}$, the global error satisfies $e_N \leq Ch^p e^L$ for constants $C, p > 0$ depending on the solver. Conditioning on this event and choosing $h$ sufficiently small yields, for any $\varepsilon > 0$, $P(e_N > \varepsilon) \leq P(L_{\text{eff}}(x_1) > L) \leq \delta$. Full convergence in probability ($\lim_{h \to 0} P(e_N > \varepsilon) = 0$ for all $\varepsilon > 0$) follows by taking $L \to \infty$ along quantiles of $L_{\text{eff}}$ and adjusting $h$ accordingly; see Appendix E for details.

Definitions C.1–C.4 are stated in terms of the exact ODE solution $\{x_t\}$, which is not directly accessible in practice. Accordingly, our experiments report $\widehat{L}_{\text{eff}}^{(h)}$ (Definition C.2) computed along numerical trajectories as a proxy. This is evidence for sensitivity and it is not, by itself, a guarantee that $L_{\text{eff}}$ is bounded for the exact flow. All convergence statements below are therefore conditional on the (unobserved) typical-set of trajectories whose effective Lipschitz constant stays bounded.

## C.2 EMPIRICAL VALIDATION OF THE CONVERGENCE ARGUMENT

We design three experiments to validate the convergence argument, each targeting one step of the logical chain: (1) bounded local error, (2) bounded error propagation via trajectory-local sensitivity, and (3) convergence under step-size refinement. Appendices F and G provide additional technical details.

### C.2.1 EXPERIMENT 1: VERIFYING BOUNDED LOCAL DISCRETIZATION ERROR

The goal of this experiment is to test that the numerical solver introduces bounded local truncation error at each step.

We measure local truncation error by comparing single-step outputs between Heun's method and a high-precision reference (Heun with $h/10$ subdivisions). We compute this at 1000 randomly selected trajectory points across 1,000 samples for each routing strategy.

The following measurements are computed: 1) mean and maximum local error $\epsilon^{\text{local}}$ per routing strategy, 2) scaling with step size at $h \in \{0.02, 0.01, 0.005\}$ for each routing strategy, and 3) distribution of local errors across timesteps and samples for each routing strategy. Detailed results are reported in Appendix G.2.

### C.2.2 EXPERIMENT 2: MEASURING TRAJECTORY-LOCAL SENSITIVITY

The goal of this experiment is to test that the trajectory exhibits bounded Jacobian spectral norms, and that this bounds error propagation. We emphasize that spectral norms quantify worst-case local sensitivity; we do not estimate Jacobian eigenvalues or logarithmic norms along trajectories, and therefore do not directly characterize linear stability of the flow. We treat this as a limitation and leave eigenvalue-based analyses to future work.

For each numerical sampling trajectory $\{\tilde{x}_{t_n}^{(h)}\}_{n=0}^N$ starting from $x_1$ we compute $\widehat{L}_{\text{eff}}^{(h)}(x_1)$. However, computing a Jacobian spectral norm can be time consuming for latent spaces of high dimension, and hence, we estimate it using the power method. We repeat the power iteration until the relative change between iterations 9 and 10 is less than $0.5\%$. For samples with slower convergence, we extend up to 20 iterations.

The following measurements are computed: 1) $\widehat{L}_{\text{eff}}^{(h)}$ distribution per routing strategy including mean and standard deviation, and 2) temporal profile of $\|J_x v(\tilde{x}_{t_n}^{(h)}, t_n)\|_2$ as a function of $t_n$. Results are reported in Appendix G.3 and Figure 2.

### C.2.3 EXPERIMENT 3: STEP-SIZE REFINEMENT AND CONVERGENCE

The goal of this experiment is to test whether step-size refinement reduces endpoint error, and whether step-refinement disagreement relates to trajectory-local stability.

For each initial noise $x_1$, we generate samples at $N$ steps (step size $h = 1/N$) and $2N$ steps (step size $h/2$). We measure the step-refinement disagreement defined as

$$\Delta_{\text{refine}}(x_1) := \text{LPIPS}\big(D(\tilde{x}_0^{(h)}), D(\tilde{x}_0^{(h/2)})\big), \tag{1}$$

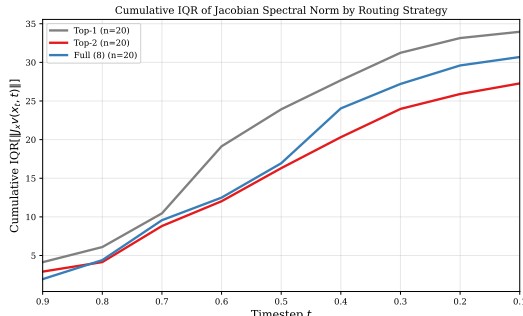

Figure 2: Cumulative IQR of the Jacobian spectral norm $\|\nabla_x v(x_t, t)\|$ as a measure of variability across sampling trajectories. The gap between Top-2 and other strategies widens as denoising progresses. Mid-trajectory timesteps ($t \in [0.1, 0.9]$); $n{=}20$ samples per strategy.

where $D(\cdot)$ denotes VAE decoding, LPIPS measures perceptual distance in decoded image space Zhang et al. (2018), and $x_1$ is the initial noise. The metric $\Delta_{\text{refine}}$ is an a posteriori refinement-based error indicator that does not involve Jacobian computation, providing a measure of numerical integration error that avoids methodological coupling with $L_{\text{eff}}$.

The following measurements are computed: 1) $\Delta_{\text{refine}}$ distribution per routing strategy, 2) Spearman correlation coefficient $\rho(\widehat{L}_{\text{eff}}^{(h)}, \Delta_{\text{refine}})$, and 3) area under the curve (AUC) for predicting high-$\Delta_{\text{refine}}$ events using $\widehat{L}_{\text{eff}}^{(h)}$ to measure predictive performance. Results are reported in Figure 3; see Appendix G.4 for methodology details.

### C.3 SUMMARY OF SENSITIVITY RESULTS

Across all routing strategies, correlation between $\widehat{L}_{\text{eff}}^{(h)}$ and step-refinement disagreement $\Delta_{\text{refine}}$ is low ($\rho < 0.08$; Figure 3)[1]. This further supports our finding that numerical stability metrics do not govern generation quality, aligning with the observed dissociation between sensitivity and FID. Factors beyond worst-case Jacobian norms—such as directional alignment of perturbations with the flow or cancellation effects across timesteps—likely govern actual error accumulation.

## D WHEN $\widehat{L}_{\text{EFF}}^{(h)}$ APPROXIMATES $L_{\text{EFF}}$

Assume the exact ODE solution $\{x_t\}_{t \in [0,1]}$ exists and remains in a set $K$, and that for all $t \in [0, 1]$, $J_x v(\cdot, t)$ is $L_J$-Lipschitz on $K$, that is,

$$\|J_x v(x, t) - J_x v(y, t)\| \le L_J \|x - y\|$$

for all $x, y \in K$. Let $\tilde{x}^{(h)}(\cdot)$ be any continuous-time interpolation of the numerical trajectory such that $\sup_{t \in [0,1]} \|x_t - \tilde{x}^{(h)}(t)\| \le \eta(h)$ with $\eta(h) \to 0$ as $h \to 0$, and define

$$\widehat{L}_{\text{eff,cont}}^{(h)}(x_1) := \sup_{t \in [0,1]} \|J_x v(\tilde{x}^{(h)}(t), t)\|.$$

Then

$$\left| L_{\text{eff}}(x_1) - \widehat{L}_{\text{eff,cont}}^{(h)}(x_1) \right| \le L_J \, \eta(h),$$

so $\widehat{L}_{\text{eff,cont}}^{(h)}(x_1) \to L_{\text{eff}}(x_1)$ as $h \to 0$. Moreover, if $t \mapsto \|J_x v(\tilde{x}^{(h)}(t), t)\|$ is $L_t$-Lipschitz on $[0, 1]$, then the grid maximum in Definition C.2 satisfies

$$0 \le \widehat{L}_{\text{eff,cont}}^{(h)}(x_1) - \widehat{L}_{\text{eff}}^{(h)}(x_1) \le L_t h,$$

so $\widehat{L}_{\text{eff}}^{(h)}(x_1)$ is a consistent proxy for $L_{\text{eff}}(x_1)$ under refinement.

This shows that the gap between this grid maximum and a continuous-time supremum is $O(h)$.

---

[1] Preliminary analysis of the router Jacobian term $|\sum_k v_k \nabla_x w_k|$ also shows similarly low correlation with $\Delta_{\text{refine}}$ (Spearman $\rho = -0.07$, $p = 0.62$ for Top-2 routing, $n{=}50$), reinforcing that Jacobian-based sensitivity metrics are not aligned with discretization error.

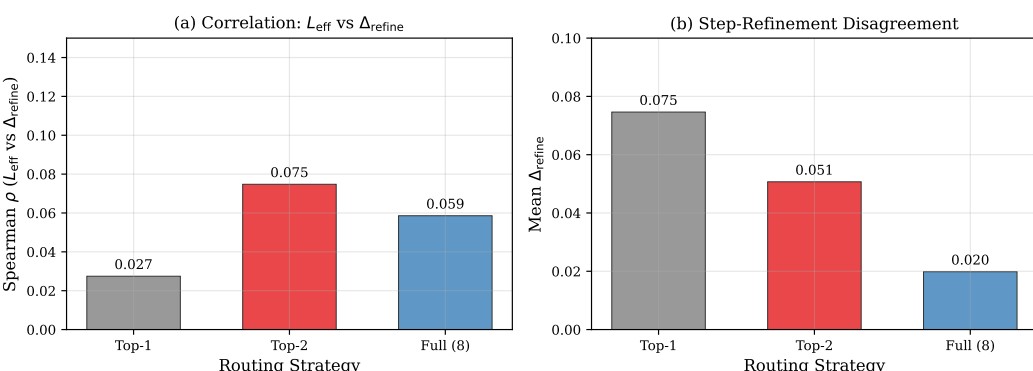

Figure 3: Correlation between trajectory sensitivity and step-refinement disagreement from Experiment 3 ($n{=}1000$ samples per routing strategy). **(a)** Spearman correlation $\rho(\widehat{L}_{\text{eff}}^{(h)}, \Delta_{\text{refine}})$ is weak across all strategies ($\rho < 0.08$), indicating that $L_{\text{eff}}$ is not a tight predictor of discretization error. **(b)** Mean step-refinement disagreement $\Delta_{\text{refine}}$ shows clear ordering: Full ensemble achieves the lowest discretization error (0.020), followed by Top-2 (0.051) and Top-1 (0.075).

## E   PROBABILISTIC CONVERGENCE UNDER EMPIRICAL STABILITY

This appendix provides the formal convergence argument sketched in Section C.1. We use a standard conditioning approach combined with deterministic ODE error bounds.

**Proposition E.1** (Conditional convergence)**.** *Let $v$ be a velocity field and $q$ a noise distribution. Suppose that for $x_1 \sim q$, the effective Lipschitz constant $L_{\text{eff}}(x_1)$ (Definition C.1) satisfies $P(L_{\text{eff}}(x_1) < \infty) = 1$. Let $\tilde{x}_{t_N}^{(h)}$ denote the numerical solution at the final step using step size $h$, and let $e_N := \|x_{t_N} - \tilde{x}_{t_N}^{(h)}\|$ be the global error. Then for any $\varepsilon > 0$,*

$$\lim_{h \to 0} P(e_N > \varepsilon) = 0.$$

*Proof.* Fix $\varepsilon > 0$ and $\eta > 0$. Define the events $A_L := \{L_{\text{eff}}(x_1) \leq L\}$ and $E := \{e_N > \varepsilon\}$. By the law of total probability,
$$P(E) \leq P(E \mid A_L) + P(A_L^c).$$

*Step 1 (Deterministic bound on $A_L$).* Standard Grönwall-based global error analysis for one-step methods (see, e.g., Hairer et al. (1993)) shows that if the velocity field has Lipschitz constant $L$ along the trajectory, then the global error satisfies

$$e_N \leq C h^p e^L$$

for constants $C > 0$ and $p \geq 1$ depending on the solver order and local truncation error bounds. On the event $A_L$, the trajectory-local Lipschitz constant is bounded by $L$, so this deterministic bound applies. Choosing $h^* = h^*(\varepsilon, L) := (\varepsilon/(Ce^L))^{1/p}$ ensures $e_N \leq \varepsilon$ on $A_L$ for all $h \leq h^*$. Thus $P(E \mid A_L) = 0$ for $h \leq h^*$.

*Step 2 (Choosing $L$).* Since $P(L_{\text{eff}}(x_1) < \infty) = 1$, for any $\eta > 0$ there exists $L = L(\eta)$ such that $P(A_L^c) = P(L_{\text{eff}}(x_1) > L) < \eta$.

*Step 3 (Two-parameter limit).* Given $\varepsilon, \eta > 0$: (i) choose $L = L(\eta)$ so that $P(A_L^c) < \eta$; (ii) choose $h \leq h^*(\varepsilon, L)$ so that $P(E \mid A_L) = 0$. Then $P(E) < \eta$. Since $\eta > 0$ was arbitrary, $\lim_{h \to 0} P(e_N > \varepsilon) = 0$. $\qquad\square$

*Remark* E.2. The key assumption is $P(L_{\text{eff}}(x_1) < \infty) = 1$, i.e., that almost all trajectories have finite effective Lipschitz constant. This is an empirical regularity condition that we validate by measuring $\widehat{L}_{\text{eff}}^{(h)}$ along numerical trajectories. The $(L, \delta)$-trajectory-locally sensitive condition (Definition C.4)

connects directly to this result: since the definition requires the bound to hold for *some* $L$ and $\delta$, we are free to choose any $\delta > 0$ and set $L = L(\delta)$ as the corresponding quantile of $L_{\text{eff}}$. For this $(L(\delta), \delta)$ pair and sufficiently small $h = h(\varepsilon, L(\delta))$, we obtain $P(e_N > \varepsilon) \leq \delta$. Since $\delta$ can be made arbitrarily small, this recovers full convergence in probability.

## F  EXPERT-DATA ALIGNMENT EXPERIMENT DETAILS

This section provides implementation details for the alignment experiments described in Section 4.

**DINOv2 embedding extraction.**   We use DINOv2-ViT-L/14 Oquab et al. (2023) to extract embeddings for both training data cluster centroids and intermediate sampling states. For intermediate states $x_t$ during sampling, we first decode through the VAE to obtain pixel-space images, then extract DINOv2 embeddings. The 8 cluster centroids were computed during DDM training using hierarchical k-means on DINOv2 embeddings of the training set.

**Cluster distance computation.**   For each routing decision at timestep $t$, we compute the Euclidean distance from the current sample's DINOv2 embedding to each of the 8 cluster centroids. The cluster rank is determined by sorting these distances (rank 1 = closest). For Top-$k$ routing, we report the minimum rank among selected experts.

**Velocity alignment computation.**   For each expert $k$ at each timestep, we compute the velocity prediction $v_k(x_t)$ and measure its cosine similarity with the blended velocity $v_{\text{blend}}(x_t) = \sum_{j \in \mathcal{S}} w_j v_j(x_t)$. This requires evaluating all 8 experts at each recorded timestep, increasing computational cost by approximately $4\times$ compared to standard Top-2 sampling.

**Statistical testing.**   Differences between selected and non-selected expert alignment scores are tested using paired $t$-tests, pairing by sample and timestep. Correlations are reported as Spearman's $\rho$ with two-tailed $p$-values.

## G  SENSITIVITY ANALYSIS DETAILS

This appendix provides additional details for the trajectory sensitivity analysis in Section C.

Table 5 reveals that the router term $\sum_k v_t^{(k)} \nabla_x w_t^{(k)}$ dominates the expert term by 2–4 orders of magnitude across all routing strategies. This dominance reflects the inherent sensitivity of softmax routing to input perturbations, a property shared by Top-2 and full ensembling alike. However, when *comparing* routing strategies, the router term provides limited discriminative signal: it is uniformly large regardless of how many experts are selected. The expert term $\sum_k w_t^{(k)} \nabla_x v_t^{(k)}$, by contrast, captures how the *weighted mixture of expert outputs* changes with input. For this reason, our trajectory-local sensitivity traces (Figure 2) report the expert-only Jacobian $\| \sum_k w_t^{(k)} \nabla_x v_t^{(k)} \|$, isolating the component that explains sensitivity differences between routing strategies. The router gradient dominance is documented separately in Figure 4 and Table 5.

### G.1  EXPERIMENTAL SETUP

We use the pretrained DDM Paris model Jiang et al. (2025) via released pretrained checkpoints. The model consists of $K = 8$ experts trained on a subset of LAION-Aesthetics Schuhmann et al. (2022). The dataset was partitioned into 8 semantic clusters via two-stage hierarchical k-means on DINOv2-ViT-L/14 embeddings. The model uses a DiT-B/2 router ($\sim$129M parameters) and 8 DiT-XL/2 experts Peebles & Xie (2023) (a modified version of the DiT-XL/2 experts with $\sim$606M parameters each, $\sim$5B total).

We compare Top-1, Top-2, and full-ensemble routing strategies on the DDM architecture McAllister et al. (2025). To avoid circularity, that is defining failure via $L_{\text{eff}}$ and then claiming $L_{\text{eff}}$ predicts failure, we use Eq.(1) as a methodologically independent error signal.

Table 4: **Local truncation error is routing-invariant.** One-step Heun local error $\epsilon_n^{\text{local}}$ at $h=0.01$ and $h=0.005$, with empirical scaling. Mean $\pm$ std. over $n=1000$ trajectories per routing strategy.

| Routing | $\epsilon^{\text{local}}$ ($h=0.01$) | $\epsilon^{\text{local}}$ ($h=0.005$) | Scaling |
|---|---|---|---|
| Top-1 | 0.539 $\pm$0.045 | 0.304 $\pm$0.034 | 1.776$\times$ |
| Top-2 | 0.542 $\pm$0.046 | 0.306 $\pm$0.034 | 1.773$\times$ |
| Full (8) | 0.543 $\pm$0.048 | 0.307 $\pm$0.035 | 1.771$\times$ |

Table 5: Jacobian decomposition at $t=0.5$ ($n=100$ samples). Both terms are measured along sampling trajectories. The router gradient term dominates for both strategies. Full ensemble shows nominally higher router term means, though the difference is small relative to variance.

| Strategy | $\|J_{\text{expert}}\|$ | $\|J_{\text{router}}\|$ | Dominant |
|---|---|---|---|
| Top-2 routing | 7.58 $\pm$1.53 | 923 $\pm$1.4K | Router |
| Full ensemble (8) | 7.58 $\pm$1.59 | 1161 $\pm$2.1K | Router |

We set $\tau_{\text{refine}}$ as the 99th percentile of $\Delta_{\text{refine}}$ on Top-2 runs, then apply this fixed threshold across all methods. This avoids tuning thresholds to match $\widehat{L}_{\text{eff}}^{(h)}$ predictions.

Appendix I presents additional experiments that test the robustness of our main findings.

### G.2 EXPERIMENT 1: LOCAL TRUNCATION ERROR

Following Section C.2.1, we estimate one-step local truncation error by comparing a single Heun step of size $h$ Karras et al. (2022) against a higher-precision reference obtained by subdividing the same interval into 10 Heun sub-steps of size $h/10$. For each routing strategy, we sample 1,000 trajectories and evaluate at randomly selected trajectory points $(x_t, t)$ the maximum local truncation error defined as $\epsilon^{\text{local}} = \max_n \|\epsilon_n^{\text{local}}(t_n)\| = Ch^{p+1}$ for some constant $C$. Table 4 reports these measurements.

Local error is essentially identical across routing strategies, confirming that routing does not affect single-step numerical accuracy.

### G.3 EXPERIMENT 2: TRAJECTORY-LOCAL SENSITIVITY

We track $\|J_x v(\tilde{x}_{t_n}^{(h)}, t_n)\|$ across time for Top-1, Top-2, and full ensemble, and separate norms for selected vs. suppressed experts.

For a routed vector field $v_t(x_t) = \sum_k w_t^{(k)}(x_t) v_t^{(k)}(x_t)$, the Jacobian satisfies

$$\nabla_x v_t = \sum_k w_t^{(k)} \nabla_x v_t^{(k)} + \sum_k v_t^{(k)} \nabla_x w_t^{(k)}.$$

This decomposition separates sensitivity from experts ($\sum_k w_t^{(k)} \nabla_x v_t^{(k)}$) from sensitivity from routing ($\sum_k v_t^{(k)} \nabla_x w_t^{(k)}$). Since $\sum_k w_k = 1$ implies $\sum_k \nabla_x w_k = 0$, the router term can be rewritten as $\sum_k (v_k - v_t) \nabla_x w_k$, showing that its magnitude is governed by inter-expert disagreement times router sensitivity—large $\|\nabla_x w_k\|$ alone does not inflate this term if experts agree. We report norms of each term separately as a diagnostic; by the triangle inequality, the full Jacobian norm satisfies $\|\nabla_x v_t\| \leq \|J_{\text{expert}}\| + \|J_{\text{router}}\|$, but these bounds need not be tight. Table 5 reports the two terms evaluated at $t=0.5$ (mean $\pm$ std. over $n=100$ trajectories).

Table 5 reveals the router term dominates by 2–4 orders of magnitude, but is similar across strategies. Since this shared dominance cannot explain the quality differences observed between routing strategies, the expert term $\sum_k w_k \nabla_x v_k$—which captures how the weighted mixture of expert outputs responds to input perturbations—is the relevant quantity for understanding routing-quality relationships. We therefore report the expert-only Jacobian in Figure 2.

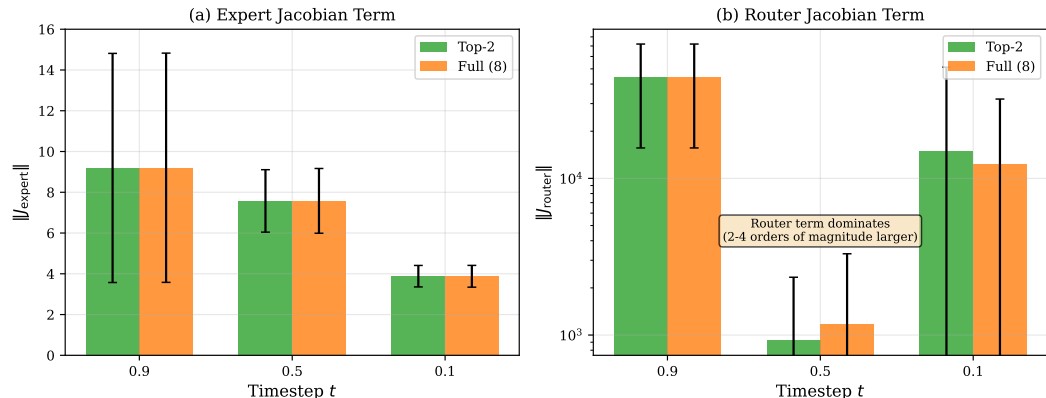

Figure 4: Temporal profile of the two Jacobian terms under Top-2 and full ensembling. Left: Expert term $\|\sum_k w_k \nabla_x v_k\|$ (linear scale). Right: Router term $\|\sum_k v_k \nabla_x w_k\|$ (log scale). The router term dominates by 2–4 orders of magnitude, but both routing strategies show similar router contributions.

### G.4   EXPERIMENT 3: STEP-SIZE REFINEMENT

We compute $\Delta_{\text{refine}}$ by running $N{=}50$ and $2N{=}100$ steps from the same initial noise $x_1$ and prompt, decoding both final latents with $D(\cdot)$, and measuring LPIPS in image space. Figure 3 shows the correlation between $\widehat{L}_{\text{eff}}^{(h)}$ and $\Delta_{\text{refine}}$.

## H   LIMITATIONS

**Correlation strength.**    The weak correlations between $L_{\text{eff}}$ and $\Delta_{\text{refine}}$ in our main in-distribution setting ($\rho < 0.1$) limit the predictive utility of trajectory Jacobian analysis for identifying failure cases. Future work should explore whether alternative sensitivity metrics (e.g., integrated sensitivity, switching frequency) provide stronger predictive signals.

**Unexplained typical-set attractivity.**    Our convergence argument relies on a trajectory-local boundedness condition ($L_{\text{eff}}(x_1) < \infty$), and our empirical results are consistent with trajectories remaining in moderate-sensitivity regions. However, we do not provide a proof that the *exact* probability flow dynamics must enter and remain in such low-sensitivity regions, nor do we characterize basins of attraction for the routed field. Developing a dynamical explanation is an important direction for future work.

**Scope.**    This paper makes a *mechanistic* claim about decentralized expert systems: routing trades off numerical sensitivity and expert-data alignment, and alignment can dominate quality. Because this claim is established by controlled comparisons that hold the expert pool and router fixed, it does not require external diffusion baselines. Adding external models without matched training would primarily answer a different question, and could obscure the routing mechanism due to unavoidable training/data confounds.

## I   EXTRA EXPERIMENTS

This subsection presents additional experiments that test the robustness of our main findings.

### I.1   FULL-ENSEMBLE TUNING VARIANTS.

We test whether the full ensemble's sensitivity metrics can be improved via inference-only modifications that keep the expert pool fixed and require no retraining: (i) temperature scaling of router logits sweeping $T \in \{0.1, 0.25, 0.5, 1.0, 2.0, 4.0\}$, and (ii) top-$p$ truncation of the router distribution, keeping the smallest set of experts whose cumulative mass exceeds $p$ and renormalizing within

Table 6: **Full-ensemble rescue attempts (no retraining).** Temperature scaling uses $w_T = \text{softmax}(z/T)$; we report the best $T$ from a sweep over $\{0.1, 0.25, 0.5, 1.0, 2.0, 4.0\}$. Top-$p$ truncation keeps the smallest set of experts whose cumulative probability exceeds $p$ and renormalizes. Results from $n{=}50$ samples per configuration.

| Variant | $\widehat{L}_{\text{eff}}^{(h)} \downarrow$ | $\Delta_{\text{refine}} \downarrow$ |
|---|---|---|
| Full ensemble (baseline) | $15.97 \pm_{5.51}$ | $0.043 \pm_{0.052}$ |
| Temp scaling ($T{=}4.0$) | $\mathbf{15.00} \pm_{4.71}$ | $0.044 \pm_{0.056}$ |
| Top-$p$ truncation ($p^\star{=}0.9$) | $15.85 \pm_{5.30}$ | $\mathbf{0.043} \pm_{0.052}$ |

Table 7: **Counterfactual routing interventions.** Mean $\Delta_{\text{refine}}$ from Eq. 1 (computed without Jacobian metrics). Results from $n{=}50$ samples.

| Condition | $\Delta_{\text{refine}} \downarrow$ |
|---|---|
| **Top-2 (base)** | $0.043 \pm_{0.054}$ |
| Full ensemble (8) | $0.039 \pm_{0.052}$ |
| Full + weight clip ($\|\nabla_x v_k\| < \text{median}$) | $\mathbf{0.037} \pm_{0.029}$ |
| Misaligned Top-2 (random) | $0.040 \pm_{0.035}$ |

this set. Table 6 summarizes the best variants from each sweep. Neither modification substantially changes $\widehat{L}_{\text{eff}}^{(h)}$ or $\Delta_{\text{refine}}$ compared to the baseline full ensemble.

## I.2 COUNTERFACTUAL ROUTING

We evaluate two inference-only counterfactuals for the full ensemble (Table 7): (i) weight clipping that suppresses experts whose Jacobian norms are above the median at the current state, and (ii) **Misaligned Top-2** (random expert selection), which preserves sparsity while explicitly breaking proximity-based alignment.

## I.3 FAILURE MODE ANALYSIS

To understand how different routing strategies fail, we categorize samples by three failure indicators: high routing uncertainty, poor numerical convergence, and high effective Lipschitz constant. Table 12 reports the frequency of each failure mode across routing strategies.

## I.4 FULL-ENSEMBLE RESCUE ATTEMPTS

We test inference-time modifications to the full ensemble to investigate whether its poor sample quality (despite superior numerical stability) can be improved without retraining. As discussed in Section C, full ensemble has the lowest $\widehat{L}_{\text{eff}}^{(h)}$ and $\Delta_{\text{refine}}$ but worst FID due to expert-data misalignment: experts process out-of-distribution inputs from data partitions they were not trained on.

We test inference-time temperature scaling of router logits: $w_T(x,t) = \text{softmax}(z(x,t)/T)$. We sweep $T \in \{0.1, 0.25, 0.5, 1.0, 2.0, 4.0\}$ for the *full ensemble* strategy ($v(x,t) = \sum_{k=1}^{K} w_{T,k}(x,t)v_k(x,t)$), holding the expert pool and solver fixed. Table 8 reports the results of the temperature sweep.

We also sweep $p \in \{0.8, 0.9, 1.0\}$. Table 9 reports the results. Truncation has minimal effect on sensitivity metrics.

## I.5 GENERALIZATION TESTS WITHOUT RETRAINING

To test whether the sensitivity ordering is specific to the training distribution, we evaluate the same frozen experts and router on a held-out prompt set. We generate $n{=}100$ samples per routing strategy

Table 8: Temperature scaling sweep for full ensemble routing. Lower temperatures sharpen the weight distribution toward top experts; higher temperatures flatten it. All metrics measured on $n=100$ samples. $T=4.0$ achieves the lowest $\Delta_{\text{refine}}$ in this sweep.

| $T$ | **Entropy** | $\widehat{L}_{\text{eff}}^{(h)}$ | $\Delta_{\text{refine}}$ |
|---|---|---|---|
| 0.10 | 0.17 | 18.84 $\pm 7.43$ | 0.060 $\pm 0.047$ |
| 0.25 | 0.43 | 18.68 $\pm 7.07$ | 0.054 $\pm 0.038$ |
| 0.50 | 0.79 | 18.06 $\pm 5.65$ | 0.054 $\pm 0.051$ |
| 1.00 | 1.24 | 17.32 $\pm 5.48$ | 0.050 $\pm 0.049$ |
| 2.00 | 1.70 | 16.35 $\pm 4.56$ | 0.047 $\pm 0.045$ |
| 4.00 | 1.96 | 16.17 $\pm 4.88$ | 0.044 $\pm 0.034$ |

Table 9: Top-$p$ truncation sweep for full ensemble routing ($T=1.0$). Lower $p$ values exclude low-probability experts. All metrics measured on $n=50$ samples. Truncation has minimal effect on $\widehat{L}_{\text{eff}}^{(h)}$ or $\Delta_{\text{refine}}$.

| $p$ | $\widehat{L}_{\text{eff}}^{(h)}$ | $\Delta_{\text{refine}}$ |
|---|---|---|
| 0.8 | 16.51 $\pm 5.64$ | 0.043 $\pm 0.052$ |
| 0.9 | 15.85 $\pm 5.30$ | 0.043 $\pm 0.052$ |
| 1.0 | 15.97 $\pm 5.51$ | 0.043 $\pm 0.052$ |

and report the same sensitivity diagnostics. Table 10 shows that the $\Delta_{\text{refine}}$ ordering (Full lowest) persists under this prompt shift, while $\widehat{L}_{\text{eff}}^{(h)}$ values are comparable across strategies.

We also evaluate two forms of distribution shift: (i) out-of-distribution prompts and (ii) stressed numerical regimes.

We evaluate the fixed trained experts and router on COCO captions as prompts. We sample $n=100$ prompts uniformly at random from the caption set, generate one sample per prompt with fixed seeds, and compute $\widehat{L}_{\text{eff}}^{(h)}$ and $\Delta_{\text{refine}}$ as in the main text. We additionally report the AUC of $\widehat{L}_{\text{eff}}^{(h)}$ for predicting high $\Delta_{\text{refine}}$ events using the same percentile thresholding protocol. We repeat the same evaluation under a harder numerical regime without retraining: (i) fewer solver steps (Heun-25 instead of Heun-50), and (ii) higher CFG Ho & Salimans (2022) (7.5 instead of 4.0). The goal is not quality, but whether sensitivity rankings and predictiveness persist. Table 13 compares baseline and stressed regimes.

## I.6 SWITCHING SENSITIVITY: MARGIN AND VECTOR-FIELD GAP

We analyze nonsmooth expert switching by combining (i) proximity to the switching surface (router margin) and (ii) the jump size in the routed vector field (vector-field gap). This addresses the limitation that for hard Top-1 the Jacobian $J_x v$ is defined only inside routing regions and does not capture discontinuities at switches.

Let $z_k(x,t)$ be router logits, $p(k \mid x,t) = \text{softmax}(z(x,t))_k$, and let $k_{(1)}, k_{(2)}$ index the top-2 logits. We report: (1) probability margin $m_p = p_{(1)} - p_{(2)}$, (2) logit margin $m_z = z_{(1)} - z_{(2)}$, (3) vector-field gap $g = \|v_{k_{(1)}} - v_{k_{(2)}}\|_2$, (4) switching score $S_{\text{switch}} = g/(m_z + \epsilon_{\text{sw}})$ with $\epsilon_{\text{sw}} = 10^{-3}$ for numerical stability. For each trajectory we summarize by $S_{\text{eff}} = \max_t S_{\text{switch}}(x(t), t)$ and an integrated variant $S_{\text{int}} = \int_0^1 S_{\text{switch}}(x(t), t)\, dt$.

For each saved trajectory state $(x_t, t)$ we compute $(m_p, m_z)$ from the router, then evaluate the two corresponding experts $v_{k_{(1)}}, v_{k_{(2)}}$ to obtain $g$ and $S_{\text{switch}}$. For Top-1, this requires one extra expert evaluation at analysis time. We subsample timesteps identically to the $L_{\text{eff}}$ pipeline.

We first reproduce margin statistics as a proxy for distance to switching surfaces. See Table 14.

Table 10: **Generalization test on COCO captions (out-of-distribution prompts).** Frozen experts/router, new prompt distribution. Results from $n{=}100$ samples with Heun solver (50 steps). $\Delta_{\text{refine}}$ ordering matches Table 1 (Full lowest), while $\widehat{L}_{\text{eff}}^{(h)}$ values are comparable across strategies.

| Strategy | $\widehat{L}_{\text{eff}}^{(h)} \downarrow$ | $\Delta_{\text{refine}} \downarrow$ | Spearman $\rho(\widehat{L}_{\text{eff}}^{(h)}, \Delta_{\text{refine}})$ |
|---|---|---|---|
| Top-1 | 27.17 ±12.08 | 0.114 ±0.109 | 0.11 |
| **Top-2** | 26.32 ±11.95 | 0.083 ±0.089 | 0.01 |
| Full (8) | 26.37 ±11.83 | **0.048** ±0.067 | −0.01 |

Table 11: Comparison of failure predictors for Top-1 routing. $S_{\text{eff}}$ (switching score) combines margin and vector-field gap. The vector-field gap $g$ alone achieves the best prediction of high $\Delta_{\text{refine}}$ (AUC 0.63).

| Top-1 predictor | AUC (high $\Delta_{\text{refine}}$) | Spearman vs. $\Delta_{\text{refine}}$ |
|---|---|---|
| $m_p$ only | 0.50 | 0.07 |
| $g$ only | **0.63** | **0.20** |
| $S_{\text{eff}}$ (margin+gap) | 0.55 | 0.08 |
| $L_{\text{eff}}$ only | 0.58 | 0.08 |
| $L_{\text{eff}} + S_{\text{eff}}$ | 0.58 | 0.11 |

Low-margin segments concentrate in failures, but margin alone does not measure the impact of switching. We evaluate how different predictors perform at identifying failures within Top-1 routing in Table 11.

The key findings are: (1) the vector-field gap $g$ alone is the best single predictor for high $\Delta_{\text{refine}}$, (2) combining $L_{\text{eff}}$ with switching features does not improve over $g$ alone.

Table 12: Failure mode analysis across routing strategies ($n{=}100$ samples). Thresholds: Routing uncert. = max entropy $>1.5$ nats; Poor conv. = $\Delta_{\text{refine}}{>}0.1$; High $L_{\text{eff}}$ = $\widehat{L}_{\text{eff}}^{(h)}{>}50$. Poor convergence decreases with more experts. High $L_{\text{eff}}$ events are rare. Note: Routing uncertainty naturally increases with ensemble size since more experts contribute non-zero weights.

| Strategy | $\widehat{L}_{\text{eff}}^{(h)}$ | $\Delta_{\text{refine}}$ | Routing uncert. | Poor conv. | High $L_{\text{eff}}$ |
|---|---|---|---|---|---|
| Top-1 | 24.21 $\pm$7.44 | 0.112 $\pm$0.114 | 60% | 39% | 1% |
| **Top-2** | 23.68 $\pm$8.07 | 0.094 $\pm$0.074 | 64% | 33% | 1% |
| Top-4 | 23.56 $\pm$6.74 | 0.049 $\pm$0.059 | 88% | 11% | 0% |
| Full (8) | 23.72 $\pm$6.97 | **0.038** $\pm$0.053 | 94% | 5% | 1% |

Table 13: Generalization under baseline vs. stressed numerical regimes (COCO captions, $n{=}100$). Stressed regime uses fewer solver steps (Heun-25) and higher CFG (7.5). Sensitivity rankings persist across regimes.

| Strategy | Regime | $\widehat{L}_{\text{eff}}^{(h)}$ | $\Delta_{\text{refine}}$ | Spearman | AUC |
|---|---|---|---|---|---|
| Top-1 | Baseline | 20.09 $\pm$12.61 | 0.131 $\pm$0.128 | 0.09 | 0.56 |
| Top-2 | Baseline | 18.39 $\pm$5.57 | 0.078 $\pm$0.079 | **0.31** | **0.69** |
| Full (8) | Baseline | **17.29** $\pm$5.82 | **0.054** $\pm$0.073 | 0.05 | 0.52 |
| Top-1 | Stressed | 18.16 $\pm$5.99 | 0.206 $\pm$0.134 | 0.20 | 0.54 |
| Top-2 | Stressed | 17.27 $\pm$5.53 | 0.190 $\pm$0.148 | 0.03 | 0.59 |
| Full (8) | Stressed | **16.07** $\pm$4.68 | **0.136** $\pm$0.135 | $-0.06$ | 0.48 |

Table 14: Router probability margin statistics for stable vs. unstable samples (unstable = $\Delta_{\text{refine}}$ above 75th percentile). Unstable samples exhibit lower margins and more frequent near-switching events, but margin alone is insufficient for prediction.

| Routing | Median $m_p$ | % Steps $m_p < 0.05$ |
|---|---|---|
| Top-1 (stable samples) | 0.42 $\pm$ 0.18 | 8.3% |
| Top-1 (unstable samples) | 0.21 $\pm$ 0.14 | 24.7% |
| Top-2 (stable samples) | 0.38 $\pm$ 0.16 | 9.1% |
| Top-2 (unstable samples) | 0.19 $\pm$ 0.12 | 27.2% |

