# OpenReview forum: "Expert-Data Alignment Governs Generation Quality in Decentralized Diffusion Models"
_ICLR.cc/2026/Workshop/Sci4DL — Sci4DL 2026_

### Official Review · Reviewer_Y3bK · 2026-02-20

**Fit:** 1
**Significance:** 2
**Confidence:** 2

**Summary:**

This paper studies decentralized diffusion models (DDMs), which combine independently trained models on disjoint datasets, and investigates what governs their generation quality. The main finding is that the key factor is the alignment between routing inputs and experts, rather than the smoothness of the denoising trajectory. The experimental results validate this claim across two systems.

Overall, I find the contribution interesting and the empirical evidence convincing. However, there are several points that require clarification before the paper is ready for publication.

**Strengths:**

- The core finding are interesting: expert-data alignment matters more than numerical stability, and sparse routing (Top-2) can even outperform a baseline trained on all data.

**Suggestions:**

**Important points**
- The background on decentralized diffusion and flow matching is too brief. In particular, the time convention is never made explicit (it is unclear whether $t=0$ corresponds to noise or data), and it is not stated clearly in the main text that the router weights are produced by a learned model. The authors should expand the background, ideally in an appendix.
- DDMs can be viewed as a form of compositional generation, where the key difference is that the composition weights are produced by a learned router conditioned on the noisy image $x_t$. Given this connection, the paper would benefit from a comparison with compositional generation methods (see [1]-[5]).
- Regarding the cluster distance computation at intermediate timesteps: the paper uses the DINOv2 embedding of the noisy image $x_t$, decoded through the VAE. It is not clear why this choice is preferred over computing the embedding of the predicted clean image $\mathbb{E}[x_0 \mid x_t]$, which would seem more meaningful for comparing inputs to training-data cluster centroids in a noise-free space.
- It is also unclear how the cluster distances computed at different timesteps ($t \in \{0.3, 0.5, 0.7\}$) are aggregated into a single cluster rank statistic. This should be described explicitly, along with a discussion of how informative the distances are at different noise levels.
- The description of $D_\text{int}$ and Figure 1 is unclear. Since $D_\text{int}$ is a single scalar per sample, it is not obvious what each quartile represents — the paper should state explicitly that samples are binned by their per-sample value of $D_\text{int}$.

**Minor points**
- The correlation value cited in the conclusion is never introduced in the main text, nor is the reader pointed to the appendix where it is derived. A forward reference should be added.


[1] Thornton, J., Béthune, L., Zhang, R., Bradley, A., Nakkiran, P., & Zhai, S.. Composition and control with distilled energy diffusion models and sequential monte carlo. AISTATS 2025

[2] Khalafi, S., Hounie, I., Ding, D., & Ribeiro, A. Composition and Alignment of Diffusion Models using Constrained Learning. In The Thirty-ninth Annual Conference on Neural Information Processing Systems 2025.

[3] Skreta, Marta, Tara Akhound-Sadegh, Viktor Ohanesian, Roberto Bondesan, Alan Aspuru-Guzik, Arnaud Doucet, Rob Brekelmans, Alexander Tong, and Kirill Neklyudov. "Feynman-Kac Correctors in Diffusion: Annealing, Guidance, and Product of Experts." In International Conference on Machine Learning, pp. 55906-55949. PMLR, 2025.

[4] Du, Yilun, Conor Durkan, Robin Strudel, Joshua B. Tenenbaum, Sander Dieleman, Rob Fergus, Jascha Sohl-Dickstein, Arnaud Doucet, and Will Sussman Grathwohl. "Reduce, reuse, recycle: Compositional generation with energy-based diffusion models and mcmc." In International conference on machine learning, pp. 8489-8510. PMLR, 2023.

[5] Liu, Hao, Tony Junze Ye, Jose Blanchet, and Nian Si. "ScoreFusion: Fusing Score-based Generative Models via Kullback–Leibler Barycenters." In The 28th International Conference on Artificial Intelligence and Statistics.

---

### Official Review · Reviewer_H9JT · 2026-02-27

**Fit:** 2
**Significance:** 2
**Confidence:** 2

**Summary:**

This paper investigates what governs generation quality in Decentralized Diffusion Models (DDMs), where independently trained experts are combined via a router at inference time. The authors first show a surprising dissociation: full ensemble routing achieves the best numerical stability (lowest trajectory sensitivity and step-refinement disagreement) but the worst generation quality. They then propose expert-data alignment as the governing principle, meaning that quality depends on routing inputs to experts whose training distribution covers the current denoising state. This is validated through cluster distance analysis, per-expert velocity alignment measurements, and expert disagreement correlations. The findings are replicated on both a LAION-trained Paris DDM and a controlled MNIST setup.

**Strengths:**

1. The central finding is clean and counterintuitive. The idea that numerical stability and generation quality can be completely dissociated in DDMs is a useful insight that challenges the natural assumption that smoother dynamics should produce better outputs. Table 1 makes this point very effectively.
2. The MNIST validation is a nice addition. The fact that the alignment effects are amplified in a setting with stronger expert specialization (43% vs. 29% angular deviation reduction) gives the results a satisfying consistency.

**Suggestions:**

The expert-data alignment finding, while well-supported, is in some sense intuitive once stated: of course routing to experts that were trained on similar data should work better than forcing all experts to process everything. The trajectory sensitivity analysis and Jacobian decompositions are thorough, but the paper stays at the level of aggregate metrics. It would help to include some qualitative comparison of what full ensemble vs. Top-2 samples actually look like, or how their trajectories differ in embedding space over the course of denoising. This would make the expert-data alignment principle more concrete and help readers build geometric intuition for why smooth velocity fields can still produce poor outputs.

---

### Meta-Review · Area_Chair_89KP · 2026-03-02

**Recommendation:** Accept

**Metareview:**

The work studies generation quality in Decentralized Diffusion Models (DDMs). It shows that numerical stability and generation quality are dissociated, and expert-data alignment is the primary determinant of generation quality. The methodology is sound, and the findings can be of interest for the workshop.

---

### Decision · Program_Chairs · 2026-03-02

Accept